# P-SpikeSSM: Harnessing Probabilistic Spiking State Space Models for Long-Range Dependency Tasks

**Malyaban Bal & Abhronil Sengupta**
School of Electrical Engineering and Computer Science
The Pennsylvania State University
University Park, PA 16802
{mjb7906,sengupta}@psu.edu

## Abstract

Spiking neural networks (SNNs) are posited as a computationally efficient and biologically plausible alternative to conventional neural architectures, with their core computational framework primarily using the leaky integrate-and-fire (LIF) neuron model. However, the limited hidden state representation of LIF neurons, characterized by a scalar membrane potential, and sequential spike generation process, poses challenges for effectively developing scalable spiking models to address long-range dependencies in sequence learning tasks. In this study, we develop a scalable probabilistic spiking learning framework for long-range dependency tasks leveraging the fundamentals of state space models. Unlike LIF neurons that rely on the deterministic Heaviside function for a sequential process of spike generation, we introduce a SpikeSampler layer that samples spikes stochastically based on an SSM-based neuronal model while allowing parallel computations. To address non-differentiability of the spiking operation and enable effective training, we also propose a surrogate function tailored for the stochastic nature of the SpikeSampler layer. To enhance inter-neuron communication, we introduce the SpikeMixer block, which integrates spikes from neuron populations in each layer. This is followed by a ClampFuse layer, incorporating a residual connection to capture complex dependencies, enabling scalability of the model. Our models attain state-of-the-art performance among SNN models across diverse long-range dependency tasks, encompassing the Long Range Arena benchmark, permuted sequential MNIST, and the Speech Command dataset and demonstrate sparse spiking pattern highlighting its computational efficiency. Our implementation source code is available at https://github.com/NeuroCompLab-psu/PSpikeSSMs.

## 1 Introduction

Spiking neural networks (SNNs) (Ghosh-Dastidar & Adeli, 2009) have garnered attention as a bio-plausible substitute for traditional artificial neural networks (ANNs). Their appeal stems from their utilization of spike-based communication among neurons, a feature that closely mimics biological processes. The inherent stateful nature of SNNs enables them to adeptly handle temporal information (Neftci et al., 2019), further enhancing their suitability for various applications (Yamazaki et al., 2022). The sparse spike-based information flow characteristic of SNNs allow event-driven computation and communication in neuromorphic hardware, leading to energy savings (Sengupta et al., 2019). SNN-based models, ideal for edge computing, have undergone rigorous testing on neuromorphic hardware platforms such as Intel Loihi 2 (Davies et al., 2021), IBM TrueNorth (Merolla et al., 2014), among others, showcasing orders of magnitude improvements in energy efficiency.

In the progression of SNN-based architecture advancements, research has predominantly focused on employing leaky-integrate-and-fire (LIF) neurons (Burkitt, 2006). While the dynamics modeled by LIF neurons are deemed biologically plausible, the actual operations within the brain entail additional layers of complexity (Hodgkin & Huxley, 1990) and stochasticity (Harrison et al., 2005) that are not fully captured by the simplified LIF neuron model. Moreover, the sequential state up-

dates and spike generation using a deterministic Heaviside function complicate the training of LIF-based SNN architectures, often requiring computationally expensive methods like backpropagation through time (BPTT) (Neftci et al., 2019). This fundamental challenge has significantly limited the adoption of SNN models, particularly for complicated sequence learning tasks involving long-range dependencies. Efficient algorithmic alternative to conventional BPTT training (Bauer et al., 2023; Shrestha & Orchard, 2018; Lin et al., 2024; Bal & Sengupta, 2022) for LIF-based spiking architectures are also being proposed, but remain unexplored for long context tasks. However, in this paper, we move beyond traditional LIF-based spike generation models to develop a computationally efficient probabilistic SNN architecture, designed to effectively tackle long-range dependency tasks in the spiking domain.

State Space Models (SSMs) have been recently employed to effectively model sequence learning tasks (Gu et al., 2021a; Gu & Dao, 2023). SSMs serve as fundamental scientific models utilized across various disciplines, notably in control theory, to articulate the behavior of dynamic systems. They offer a streamlined and robust framework, facilitating a comprehensive analysis and deep understanding of the dynamics of complex systems as they unfold over time. In this work, we employ SSMs to capture temporal dependencies within sequences of input spikes, rather than conventional real-valued data. This approach not only enables computational efficiency but also underscores the remarkable capability of SSMs in analyzing long-term temporal dependencies in spike-based data.

**Probabilistic Spiking State-Space Model:** In this paper, we propose an SNN architecture grounded in a probabilistic state-space neuronal model, which we call **P-SpikeSSM**. We conceptualize the $n$-dimensional hidden state of the underlying SSM as the membrane potential, providing richer representations when compared to the scalar hidden state of an LIF neuron. Dynamics of each neuron is governed by an independent set of parameters, allowing the model to flexibly learn diverse temporal dependencies across neurons, thus enhancing its processing capacity. As outlined in the methodology, instead of real-valued inputs, we feed sequence of spikes into the P-SpikeSSM neuronal model. This enables developing a computationally efficient framework by applying convolution over the sparse spikes, instead of real-valued data. The **SpikeSampler** layer samples spikes from each P-SpikeSSM neuron, enabling parallel operation with minimal overhead. Moreover, to address the challenge of non-differentiability inherent in the stochastic spiking function, we introduce a novel **surrogate** in the form of $\mathbb{E}[S_t]$, where the discrete Bernoulli random variable $S_t$ is associated with the spiking event of each neuron at time $t$.

**Scalable Architecture with SpikeSampler and SpikeMixer:** Although individual P-SpikeSSM neurons can process one input spike sequence, addressing tasks with complex long-range dependencies demands a deeper, more scalable architecture capable of capturing diverse dependencies. To address this, we introduce a robust architecture (Fig. 1) and training framework. We encode the real-valued input sequence, associated with a sequence learning task, into $N$ distinct spike trains, which are fed to a layer consisting of $N$ corresponding neurons (see Section 3.2). Each neuron generates spikes stochastically based on its individual input spike train, while the collective activity of the neuron population allows for effectively capturing a diverse range of dependencies across the different input spike sequences. The output spikes from each neuron population in a layer are processed through a **SpikeMixer** layer, facilitating inter-neuron communication. Next, a **FuseClamp** layer performs further aggregation and computes the probability necessary for generating the subsequent spike sequences, which are then passed to the next layer of P-SpikeSSM based neuronal units. Furthermore, because the model communicates and uses sparse spike trains for computation, it achieves substantial computational efficiency by significantly reducing the number of floating-point multiplication and accumulation (MAC) operations across all layers and using simpler accumulative (ACC) operations instead. Furthermore, current state-of-the-art transformer-based spiking architectures, like (Zhou et al., 2022; Bal & Sengupta, 2024; Bal et al., 2024), process input sequences in parallel but require additional time steps to simulate network dynamics. In contrast, our proposed model eliminates this overhead, substantially improving computational efficiency.

**Application to Long-Range Dependency Tasks:** We evaluate the performance of our proposed spiking architecture on various datasets within the Long Range Arena (LRA) benchmark, along with sequence-based datasets such as permuted sequential MNIST (psMNIST) and the raw inputs of the SC10 subset of the Speech Command dataset. Our model outperforms traditional non-spiking transformer-based architectures and, to the best of our knowledge, establishes a new benchmark for spiking models in the domain of long-range arena tasks.

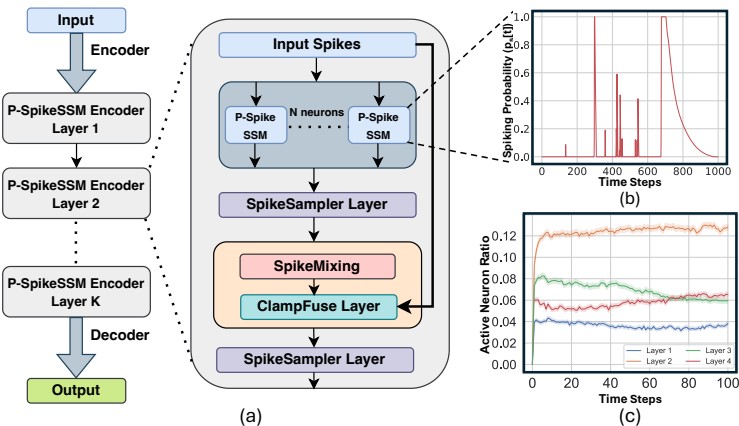

Figure 1: (a) High-level overview of the P-SpikeSSM-based spiking architecture for LRA tasks. (b) Graph depicting the sparsity of spiking events generated by a single P-SpikeSSM neuron over input sequence length (for ListOps dataset). (c) Graph showing the layer-wise active neuron ratio (i.e., the proportion of neurons generating spikes within a layer per time step) against operating time steps, for randomly sampled input from ListOps dataset. The layer-wise spiking behavior illustrates the model-wide sparsity in spiking activity, contributing to computational efficiency.

## 2    RELATED WORKS

The realm of sequence modeling is primarily dominated by transformer-based architectures. Efficient implementations like LinFormer (Wang et al., 2020), Performer (Choromanski et al., 2020), among others, have demonstrated scalability to long sequence lengths. Meanwhile, non-spiking architectures based on SSMs, such as S4 and Mamba (Gu & Dao, 2023; Gu et al., 2021a; 2020a), have also shown the capability to handle lengthy sequences. Sequence learning in SNN-based architectures have primarily been applied to vision-based datasets (Zhou et al., 2022; Fang et al., 2024) and NLP datasets (Bal & Sengupta, 2024; Zhu et al., 2023), typically with constrained sequence lengths. However, within the domain of SNNs, frameworks based on Legendre Memory Units (LMUs) (Liu et al., 2024; Voelker et al., 2020) has ventured into exploring long-range dependency tasks.

Previous efforts integrating SSMs within spiking models (Stan & Rhodes, 2023; Du et al., 2024) have primarily focused on passing the SSM output through a layer of LIF neurons to generate spikes. Using non-linear LIF neurons negates the parallel training efficiency of SSMs, as LIF neurons process information sequentially, introducing a bottleneck (see Section 3.1.4). Stan & Rhodes (2023) seeks to enhance efficiency by linearizing LIF neurons. However, because the inputs to their SSM model remain real-valued, leading to additional floating-point MAC operations, this approach fails to leverage the energy-saving potential of SNNs. Moreover, the work lacks an analysis of computational efficiency and energy benefits, particularly concerning neuron firing activity, leaving a critical aspect of model efficiency unaddressed. Furthermore, from a sequence processing standpoint, the inherent ability and use of SSMs to capture temporal dependencies renders the addition of LIF neurons superfluous, introducing unnecessary computational overhead without providing any functional improvements beyond enabling spike generation.

## 3    METHODOLOGY

In this section, we first delve into the dynamics of the proposed Probabilistic Spiking State Space Models (P-SpikeSSM). We then delve into the specifics of our proposed spiking architecture, highlighting the SpikeSampler, SpikeMixer, and FuseClamp layers. Additionally, we offer insights on scaling the P-SpikeSSM-based spiking model for tackling complex long-range dependency tasks and develop a computationally efficient parallel training framework.

### 3.1 P-SpikeSSM Formulation

We formulate the neuronal model as a time invariant system which takes in sequence of input spikes given as $x_s(t) \in \{0, 1\}$, at time $t$. Much like the membrane potential upholds the state of the LIF neuron, we anchor our approach in SSMs (Gu & Dao, 2023; Gu et al., 2021a), crafting an $n$-dimensional hidden state ($h(t)$) at time $t$. Expanding the dimensionality of the hidden state enables our neuronal model to achieve a more comprehensive state encoding of the underlying input sequence, surpassing the limitations imposed by the scalar hidden state in LIF models. The event of spike generation at time $t$ is associated with a Bernoulli random variable $S_t$ corresponding to each neuron. The probability of spiking, i.e., $p_s(t)$ at time $t$, is modeled as a function of the output of the neural model. The continuous time neuronal dynamics are expressed as,

$$
\begin{aligned}
\dot{h}(t) &= Ah(t) + Bx_s(t) \\
p_s(t) &= \sigma(Ch(t) + Dx_s(t)) \\
\sigma(z) &= clamp(az + b)
\end{aligned}
\tag{1}
$$

where, $\dot{h}(t) = \frac{dh}{dt}$ and $clamp(y) = \begin{cases} 0 & \text{if } y < 0 \\ y & \text{if } 0 \le y \le 1 \\ 1 & \text{if } y > 1 \end{cases}$, $a$ and $b$ are parameters. Setting $a = 1$

and $b = 0$ allows using the output of the SSM directly as the probability of spiking event without further scaling or translation. $A$ is a matrix controlling the evolution of the hidden state over time without any input spikes. $B$ represents the influence of the input spikes ($x_s(t)$). $C$ describes the mapping of the hidden state vector $h(t)$ to the observed outputs, i.e., $p_s(t)$. $D$ is the feedforward parameter, representing any direct influence of the inputs spikes $x_s(t)$ on the observed output probability $p_s(t)$. For the purpose of simpler formulation, following previous works on SSMs (Gu & Dao, 2023), we will consider $D = 0$, since the term $Dx_s(t)$ can be viewed as a simple skip-connection. Furthermore, $\sigma$ is a function that clamps the output between $[0, 1]$, since probability $p_s[t] \in [0, 1]$. We utilize $p_s[t]$ to sample spikes from the underlying neuron, as discussed in Section 3.1.3. The aforementioned formulation of our SSM-based neuronal model is presented in a continuous-time setting. However, since our primary focus is on sequence modeling tasks in domains such as NLP and vision tasks, we now proceed to formulate the dynamics of our neuronal model in discrete time.

#### 3.1.1 P-SpikeSSM Discrete Time Dynamics

In order to discretize our system, we sample a sequence of spikes of length $L$, given by $X_s = (x_s[1], x_s[2], ..., x_s[L])$ from the original continuous signal given by $x_s(t)$, with step size $\Delta$ such as $x_s[i] = x_s(i\Delta)$. The P-SpikeSSM neuronal dynamics are subsequently discretized using bilinear transformations (Tustin, 1947), whereby we approximate the parameters $A, B, C$ as $\overline{A}, \overline{B}, \overline{C}$ which is given as,

$$
\begin{aligned}
\overline{A} &= (I - \Delta/2 \cdot A)^{-1}(I + \Delta/2 \cdot A) \\
\overline{B} &= (I - \Delta/2 \cdot A)^{-1}\Delta B \\
\overline{C} &= C
\end{aligned}
\tag{2}
$$

where, $I$ is the Identity matrix. The transition dynamics of the discretized system at time step $t$ is,

$$
\begin{aligned}
h[t] &= \overline{A}h[t-1] + \overline{B}x_s[t] \\
p_s[t] &= \sigma(\overline{C}h[t])
\end{aligned}
\tag{3}
$$

where, $h[t]$ is the hidden state of the neuron, $p_s[t]$ is the probability of the event $S_t$. This allows us to write the transition dynamics of the system as a recurrence in discrete time. The sparse spiking dynamics of our proposed neuronal model, characterized by the spike probability $p_s[t]$, is illustrated in Fig. 1b. The spiking activity manifests in temporally localized patterns, mirroring the firing patterns observed in biological neurons (Hubel & Wiesel, 1962), with periods of non-activity interspersed between bursts. Now, instead of using a recurrent representation, we investigate how the evolution of

state dynamics can be represented by a convolution operation (with spikes as input signal), thus requiring only accumulation-based computationally efficient operation. Moreover, using convolution instead of a recurrence-based approach enables us to parallelize the framework.

### 3.1.2 REPRESENTING DYNAMICS AS CONVOLUTION OVER SPIKES

There are two problems with Eqn. 3, concerning the training of a scalable spiking architecture. Firstly, training it in its recurrent form necessitates employing a BPTT approach (Gu et al., 2021b), which is impractical for longer sequence lengths due to its time and memory overhead. Secondly, as the hidden state $h[t]$ at time $t$ can be a vector of floating points rather than spikes, the transition operations involved would not take complete advantage of energy/power efficient neuromorphic hardware. To achieve a fully parallelizable training procedure and leverage SNN-based operational efficiency during inference, we investigate an alternative formulation of Eqn. 3 as a convolution operation (Gu et al., 2021b). Since the proposed neuronal model is a time invariant system, considering the initial hidden state i.e., $h[0]$ to be a 0-vector, the recurrent relationship can be unrolled as,

$$h[i] = \overline{A}^{i-1}\overline{B}x_s[1] + \overline{A}^{i-2}\overline{B}x_s[2] + \cdots + \overline{AB}x_s[i-1] + \overline{B}x_s[i] = \sum_{j=1}^{i}(\overline{A}^{i-j}\overline{B}x_s[j]) \quad (4)$$

Thus, generalizing it to the entire sequence of length $L$ we get,

$$H = \hat{K} * X_s, \quad \hat{K} = (\overline{B}, \overline{AB}, \ldots, \overline{A}^{L-1}\overline{B}) \quad (5)$$

where, $*$ represents the non-circular convolution operation. $H$ represents the sequence of hidden states $(h[1], h[2], \ldots, h[L])$ of length $L$. $\hat{K}$ is a convolutional kernel of length $L$ as defined above (see Appendix A for further explanation). The output of the neuronal model, i.e. probability of spiking of the neuron at time $t$, is given as,

$$p_s[i] = \sigma((K * X_s)_i) \quad (6)$$

where, kernel $K = \overline{C}\hat{K} = (\overline{CB}, \overline{CAB}, \ldots, \overline{CA}^{L-1}\overline{B})$; $(K * X_s)_i = \sum_{j=1}^{i} K_j x_s[i-j+1]$ is the $i^{th}$ term of the non-circular convolution, where $K_i = \overline{CA}^{i-1}\overline{B}$. $P_s = (p_s[1], p_s[2], \ldots, p_s[L])$ is the sequence of probability of spikes from a neuron over the operating time steps. Thus, we can compute the output sequence parally by doing convolution of the input sequence of spikes with the weights of kernel $K$. Additionally, each element of the sequence $P_s$ can be computed as a dot product of a vector of real values (elements of precomputed $K$) and vector of spikes (subsequence of $X_s$). Sparse input spikes further enables skipping unnecessary computations on zero elements within the input signal. Specialized neuromorphic hardware accelerators (Ivanov et al., 2022; Davies et al., 2021) can be leveraged to perform this process, thus avoiding floating point MAC operations during inference.

### 3.1.3 SPIKESAMPLER LAYER

The spiking event of a specific P-SpikeSSM neuron at time $t$ is modeled by a Bernoulli random variable $S_t$. The spike generation process utilizes the output of the neuron, $p_s[t]$ (Eqn. 3 & 6), as the probability of spiking at time $t$, as demonstrated below:

$$S_t = \begin{cases} 1 & \text{if } z < p_s[t], \\ 0 & \text{otherwise,} \end{cases} \quad (7)$$
$$z \sim \mathcal{U}(0, 1)$$

This seemingly simple sampling process allows the proposed framework to operate in parallel without incurring significant computational overhead. This SpikeSampling process can be parallelized across both the sequence dimension and the population of neurons $N$, to develop a SpikeSampler

layer (Fig. 1a), facilitating the scaling of this methodology for more complex long-range dependency tasks. Additionally, the on-chip deployment of models that use random number sampling from a uniform distribution, on neuromorphic hardware such as Loihi-2, has been previously explored (Pierro et al., 2024). This enables the implementation of our SpikeSampler layer directly on-chip.

**Surrogate Gradient:** The stochastic nature of the SpikeSampler layer introduces challenges during training, as P-SpikeSSM-based spiking architectures face the problem of non-differentiability of spikes. To address this and enhance stability of the learning process, during the backward phase of backpropagation, we use a surrogate operation $(\overline{S}_t)$ for the stochastic spiking operation at time $t$ as,

$$\overline{S}_t = \mathbb{E}[S_t] = 0 \cdot P(S_t = 0) + 1 \cdot P(S_t = 1) = p_s[t] \tag{8}$$

### 3.1.4 WHY CHOOSE STOCHASTIC SPIKE GENERATION OVER USING LIF NEURONS?

The continuous-time internal dynamics of a basic LIF neuron is outlined below,

$$\tau_m \cdot \frac{du}{dt} = -(u(t) - u_{rest}) + R \cdot I(t) \tag{9}$$

where, at time $t$, $u(t) \in \mathbb{R}$ is the membrane potential which can be considered as the hidden state of the system; $I(t)$ is the input current scaled by a constant $R$; $\tau_m$ is the time constant associated with the decay in membrane potential over time; $u_{rest}$ is the resting membrane potential. LIF neurons thus sequentially updates its state ($u$) and uses deterministic Heaviside function for spike generation.

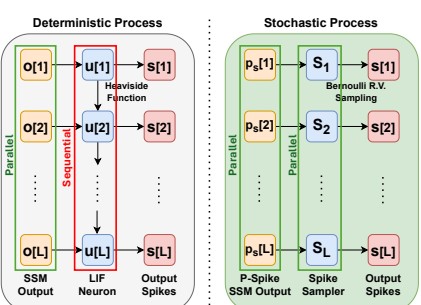

Figure 2: Computational flow of the LIF-based SSM model compared to the SpikeSampler-driven P-SpikeSSM neuronal model. Here, $L$ is the sequence length, $u[t]$ represents the membrane potential of LIF neuron and $s[t]$ denotes the spike output (either 1 or 0) at time $t$. Unlike the LIF-based approach, which is constrained by a sequential bottleneck, our probabilistic approach supports parallel processing.

**Sequential Bottleneck Issues for LIF Neurons:** The above sequential process of state update and spike generation causes a bottleneck during the parallel training of the underlying SSM based framework. This results in increase in both training and inference time (see Section 4.1.2) compared to our method. Prior work (Stan & Rhodes, 2023) attempts to linearize LIF neurons for parallel operation with SSMs, but offers limited analysis on the impact of this parallelization on model performance.

In contrast, as shown in our results, our approach not only surpasses the accuracy achieved by previous methods across multiple datasets, but does so by being computationally simpler than the former. Additionally, the previous study has not provided evidence for any contributions of LIF neurons to model performance improvement, beyond their use in spike generation.

**Parallel Execution in Our Model:** The SpikeSampler layer compliments the parallel computational advantages of the P-SpikeSSM neuron (Eqn. 6), resulting in a parallel and efficient spike generation process without the additional overhead of an LIF neuron (Fig. 2). During the backward phase, the surrogate gradient (Eqn. 8) is utilized for effective and efficient model training. To efficiently compute the kernel $K$, we capitalize on prior theoretical structural findings regarding non-spiking SSMs (Gu et al., 2021a). By exploiting the decomposition of matrix $A$ into a sum comprising a normal and low-rank matrix, we achieve efficient computation of $K$ in $O(L)$ time complexity. More details regarding efficient computation of $K$ is added in Appendix B.

### 3.2 SCALING P-SPIKESSMS TO DEEPER SNN ARCHITECTURES

To expand the sequence learning capabilities of the P-SpikeSSM neuronal model and facilitate its scalability to deeper architectures, we introduce the P-SpikeSSM neuronal layer. This layer comprises of $N$ P-SpikeSSM neurons. The sparse spiking activity of layer $i$ at time $t$ can be characterized by analyzing the active neuron ratio, denoted as $anr_i[t]$. This ratio is determined by the number of

spikes occurring in that layer at time $t$, and is defined as: $anr_i[t] = \frac{\sum_{j=1}^{N_i} s_{ij}[t]}{N_i}$, where $N_i$ is the total number of neurons in layer $i$, and $s_{ij}[t]$ represents the spike generated by neuron $j$ in that layer, at time $t$. The layer-wise sparse spiking activity of the P-SpikeSSM neurons is illustrated in Fig. 1c. The high-level architecture of the proposed spiking architecture features $K$ stacked P-SpikeSSM Encoder layers, each of which encapsulates the layers as shown in Fig. 1a. The effect of the number of neurons on model performance is shown in Fig. 3. This demonstrates that as the population of neurons within a single layer increases, the spikes generated by them more effectively capture the temporal dependencies in the input sequence.

**SpikeMixer Block:** Since each P-SpikeSSM neuron independently processes input sequence tokens, a neuron mixer layer in the form of a fully-connected block is introduced. This facilitates the aggregation of spikes from previous layer of P-SpikeSSM neurons and flow of information among neuronal layers, ensuring efficient processing of diverse temporal dependencies learned by various neurons. The output of the SpikeMixer layer is given as,

$$f_{mix}[t] = gelu(I_s[t] \cdot W_m) \qquad (10)$$

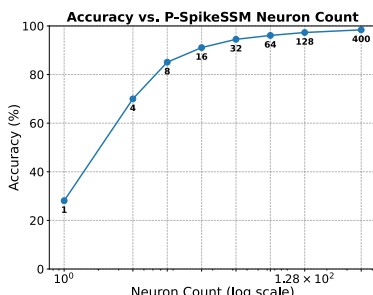

where, $I_s[t] \in \{0,1\}^N$ are the $N$ spikes from the previous SpikeSampler layer at time $t$ and $W_m \in \mathbb{R}^{N \times N}$ is a linear weight. We use $gelu$ function as a non-linearity. The linear layer in this module avoids floating-point MAC operations since the input to the FC block consists of spikes. However, due to the $gelu()$ activation, there is element-wise floating-point multiplication of $O(n^2)$ complexity (Hendrycks & Gimpel, 2016), which is still lower than the $O(n^3)$ MAC operation in floating-point matrix multiplications.

Figure 3: Results obtained from the test set of the ps-MNIST dataset. This experiment utilizes two P-SpikeSSM neuronal layers, with each layer containing $N$ neurons, represented on the x-axis. The accuracy achieved is displayed on the y-axis.

**FuseClamp Block:** The FuseClamp block combines the input spikes $(x_s)$ to the encoder layer with the SpikeMixer output via a residual connection as shown in Fig. 1a. This is followed by batch normalization $(BN)$, after which the output is clamped between $[0, 1]$ using $\sigma$. The FuseClamp block is succeeded by a SpikeSampler layer that utilizes its output as probabilities to generate spikes, which are then propagated to the subsequent layer. The operation is defined as:

$$p_{fc_i}[t] = \sigma(BN(f_{mix}[t] + x_s[t])) \qquad (11)$$

**Input Encoding and Output Decoding:** We typically employ a linear layer as the input encoder, with weight $W_e \in \mathbb{R}^{1 \times N}$, followed by an optional batch normalization layer and SpikeSampler layer. This process facilitates the generation of input spike sequences for all $N$ neurons in the subsequent layer. This choice is informed by the prevalent structure of our datasets, wherein input data sequences are commonly formatted as $\mathbb{R}^{L \times 1}$. For the long-range dependency based classification tasks, the output of the final P-SpikeSSM encoder layer, is passed through a pooling-based sequence decoder to generate the model prediction.

## 4 EXPERIMENTATION

In this section, we showcase the efficacy of our proposed P-SpikeSSM neuronal model-based SNN architecture by evaluating their performance across various long-range dependency based datasets. We also conduct a preliminary energy analysis to highlight the computational efficacy of our proposed framework. The experiments were run on Nvidia RTX A5000 GPUs (8) each with 24GB memory.

| Model | SNN | Acc. |
|---|---|---|
| S4 (Gu et al., 2021a) | No | 98.7 |
| LSTM (Gu et al., 2020b) | No | 95.1 |
| HSLMU (Voelker et al., 2020) | Yes | 96.8 |
| LMU (Voelker et al., 2019) | No | 97.2 |
| DSD-SNN (Han et al., 2023) | Yes | 97.3 |
| Transformer (Vaswani et al., 2017) | No | 97.9 |
| Spiking LMUFormer (Liu et al., 2024) | Yes* | 97.9 |
| **P-SpikeSSM (Our Model)** | **Yes\*** | **98.4** |

Table 1: Results comparing the accuracy of our model to other methods on test set of psMNIST dataset.

| Model | SNN | ListOps | Text | Retrieval | Image | Pathfinder |
|---|---|---|---|---|---|---|
| S4 (Original) (Gu et al., 2021a) | No | 58.35 | 76.02 | 87.09 | 87.26 | 86.05 |
| S4 (Improved) (Gu et al., 2021a) | No | 59.60 | 86.82 | 90.90 | 88.65 | 94.20 |
| Transformer (Vaswani et al., 2017) | No | 36.37 | 64.27 | 57.46 | 42.44 | 71.40 |
| Sparse Transformer(Tay et al., 2020) | No | 17.07 | 63.58 | 59.59 | 44.24 | 71.71 |
| Linformer (Wang et al., 2020) | No | 35.70 | 53.94 | 52.27 | 38.56 | 76.34 |
| Linear Transformer (Tay et al., 2020) | No | 16.13 | 65.90 | 53.09 | 42.34 | 75.30 |
| FLASH-quad (Hua et al., 2022) | No | 42.20 | 64.10 | 83.00 | 48.30 | 63.28 |
| Spiking LMUFormer (Liu et al., 2024) | Yes[*] | 37.30 | 65.80 | 79.76 | 55.65 | 72.68 |
| Transnormer T2 (Qin et al., 2022) | No | 41.60 | 72.20 | 83.82 | 49.60 | 76.80 |
| BinaryS4D (Stan & Rhodes, 2023) | Partial[**] | 54.80 | 82.50 | 85.30 | 82.00 | 82.60 |
| **P-SpikeSSM (Our Model)** | **Yes[*]** | **58.20** | **81.20** | **88.53** | **82.40** | **84.80** |

Table 2: Results comparing the accuracy of our model against some spiking and non-spiking architectures on test sets of LRA benchmark tasks (*Model uses $gelu$ activation but no floating point matrix multiplications, **Model uses $gelu$ act. as well as floating point matrix multiplications).

## 4.1 RESULTS

We evaluate our model (Tables 1, 2 & 3) on multiple long-range dependency tasks across datasets such as psMNIST (Le et al., 2015), Speech Command (SC10) (Warden, 2018) and the long-range arena (LRA) benchmark (Tay et al., 2020). The dataset details are in Appendix C.

**Permuted Sequential MNIST:** A simple model comprising two P-SpikeSSM Encoder layers, each with a P-SpikeSSM neuronal layer consisting of 400 neurons, achieves state-of-the-art results among spiking architectures. It performs comparably to the current best non-spiking model and outperforms non-spiking transformer-based architectures, as detailed in Table 1.

**Speech Command:** Our SNN model, featuring four P-SpikeSSM Encoder layers, each with 256 neurons, surpasses the performance of many contemporary non-spiking architectures on Speech Command 10 subset (SC10) as demonstrated in Table 3. Furthermore, on the 35-set Speech Command dataset, it achieves 96.23% outperforming previous spiking baselines (Liu et al., 2024; Bittar & Garner, 2022).

| Model | SNN | Acc. |
|---|---|---|
| S4 (Gu et al., 2021a) | No | 98.3 |
| Transformer (Vaswani et al., 2017) | No | × |
| NRDE (Gu et al., 2021a) | No | 16.5 |
| Performer (Choromanski et al., 2020) | No | 30.8 |
| CKConv(Gu et al., 2021a) | No | 71.7 |
| **P-SpikeSSM (Our Model)** | **Yes[*]** | **95.6** |

Table 3: Results comparing the accuracy obtained by our model to other non-spiking architectures on test set of SC10 dataset.

**Long Range Arena Benchmark:**

Transformer-based non-spiking models, as demonstrated in Table 2, struggle with suboptimal performance on long-context tasks in LRA benchmark. This is primarily due to the overhead incurred during the computation of attention scores, which becomes more pronounced with longer sequence lengths. In our analysis, we also compare our method against the LMU-based spiking model, SpikingLMUFormer, as well as the BinaryS4D model (Stan & Rhodes, 2023). Although BinaryS4D is not a fully spiking model (requires floating point MAC based matrix multiplications), it incorporates a layer of LIF neurons to generate spikes from an underlying state-space model (SSM).

### 4.1.1 HYPER-PARAMETERS

Initializing the state matrix $A$ with HiPPO matrices (Gu et al., 2020a) leads to optimal performance and rapid convergence. Across a majority of tasks, utilizing HiPPO-legS (further discussed in Appendix B) consistently yields the highest accuracy. Hyper-parameters employed for training our model can be found in Table 4, with additional details in Appendix D.

| Hyper-parameters | Range | Optimal |
|---|---|---|
| $K$: Encoder Layers | (2-6) | 4 |
| $N$: Neurons per Layer | (64-400) | 256 |
| $n$: Hidden State Dim. | (4-64) | 16 |
| $lr$: Learning Rate | (1e-4 - 1e-1) | 0.005 |
| Batch Size | (8-256) | 32 |
| Epochs | 20-200 | 200 |

Table 4: Hyper-parameters of our SNN models. Optimal values for ListOps dataset is also shown.

### 4.1.2 ABLATION STUDIES

**Effect of Components:** We perform experiments to analyze the effect of various components introduced in this work, specifically the surrogate (used during training) for the Spike-Sampler layer, the SpikeMixer, and use of normalization in ClampFuse layer. On more challenging datasets, such as ListOps, the model fails to achieve better-than-random accuracy when trained without the surrogate. For the LRA Text task, although training without the surrogate is feasible, it results in significantly reduced performance, as shown in Table 5. We

| Our Model | Accuracy |
|---|---|
| w/o Surrogate ($S_t$) | 70.90 |
| w/o SpikeMixer | 68.90 |
| w/o Normalization | 77.80 |
| w/ Fixed Param. $\sigma$ | 80.20 |
| w/ Learnable Param. $\sigma$ | 81.20 |
| w/ $relu$ Activation | 80.40 |

Table 5: Results showing the effect of different components of our proposed SNN architecture on test accuracy when trained on LRA Text dataset.

also ran experiments to understand the effects of SpikeMixer layer and emphasize the performance benefits of incorporating the SpikeMixer layer, which facilitates inter-neuron communication and enhances the ability of the model to capture complex temporal dependencies. Since normalization (used in the ClampFuse layer) is typically treated as a non-local operation that poses challenges for implementation on neuromorphic hardware, we conducted an experiment to assess the impact of removing normalization, and achieve promising result with minimal accuracy drop. We conducted an experiment replacing the $gelu$ layer in the SpikeMixer block with a hardware friendly $relu$ activation function (Timcheck et al., 2023), and observed minimal performance degradation.

**Parameterized $\sigma$:** The function $\sigma$ is defined with parameters $a$ and $b$ (Eqn. 1). These parameters can either be treated as hyper-parameters or learned during training. We conducted additional experiments on the LRA text dataset to analyze the impact of learning $\sigma$. Our results (Table 5) indicate that allowing $\sigma$ to be learnable improves model performance, though it introduces an additional computational cost of $O(N)$ element-wise multiplications per layer (with $N$ neurons), compared to using fixed values of $a = 0$ and $b = 1$ (i.e. using output of SSM directly for sampling probability).

**Comparing our Stochastic Model to LIF-based Implementation:** We implemented a version of our model based on LIF neurons, where the output of the underlying SSM (operating over spike sequences) is passed to an LIF neuron for spike generation. We use similar experimental setup as discussed in Section 4.1 for the psMNIST dataset. This approach achieves an accuracy of $97.7\%$ on the ps-MNIST dataset, compared to $98.4\%$ by our stochastic approach. Furthermore, our model demonstrates significantly reduced training and inference times. Specifically, training for one epoch takes approximately 1.5 minutes, compared to 6.1 minutes for the LIF-based approach on the ps-MNIST dataset. For inference, our model takes only 9 seconds on the test set, whereas the LIF-based approach requires 17 seconds. This disparity is largely due to the sequential processing bottleneck inherent in the LIF-based approach, which also necessitates Backpropagation Through Time (BPTT) (Neftci et al., 2019) during training. In contrast, our parallel framework allows for training in a single pass using standard backpropagation.

## 4.2 ANALYSIS OF ENERGY EFFICIENCY

We perform a preliminary analysis comparing the energy efficiency of a non-spiking S4 model to our spiking model during parallel inference on a 45nm CMOS technology (Han et al., 2015). For 32-bit floating points, ACC operations (cost $.9pJ$) consume $5.1\times$ less energy than MAC operations (cost $4.6pJ$) (Han et al., 2015). Assuming input sequence length of $L$, with $N$ neurons per layer across $K$ layers, the dominant computational cost per layer for the non-spiking S4 model (Gu et al., 2021a) is $NL^2$ and $LN^2$ floating point MAC operations, representing the cost of computation (underlying SSM is operated parallely and convolutional kernel is precomputed and cached) in a single S4 layer and following linear layer with hidden dimension $N$. For simplicity, we estimate the energy costs using standard convolution instead of FFT, as implementing FFT on a neuromorphic chip—which primarily relies on spike-based accumulative operations—is significantly more complex. Although a complete energy calculation includes non-linear layers like $gelu()$, their contribution is less ($O(LN)$ operations) compared to the energy cost of the parallel SSM and linear layers. Furthermore, we also explore a model variant replacing $gelu$ with $relu$ (Table 5).

Building on our previous analysis, the primary computational cost per P-SpikeSSM Encoder Layer in our spiking model is given by $IFR_{\text{in}} \cdot L^2 N$ and $IFR_{\text{o}} \cdot LN^2$ floating-point accumulation (ACC) operations, contributed primarily by the parallel operation of all the neurons of the P-SpikeSSM layer and SpikeMixer layer respectively. Here, $IFR_{\text{in}}$ represents the firing rate of the input layer to the P-SpikeSSM neuronal layer, while $IFR_{\text{o}}$ denotes the firing rate of spikes sampled from the P-SpikeSSM neuronal layer, i.e. input to the SpikeMixer. $N$ here denotes the number of neurons in a P-SpikeSSM encoder layer. As illustrated in Fig. 4, the majority of neurons in the layer remain dormant during the input sequence, leading to sparse communication.

To illustrate with a specific example, let us consider the ListOps dataset. An iso-parametric state-of-the-art non-spiking S4 model achieves an accuracy of 58.35 (improved version: 59.60) (Gu et al., 2021a), while our P-SpikeSSM achieves 58.20. For ListOps dataset, we use $L = 2K$ and hidden dimension of S4, i.e., $N = 256$. The energy consumption of the non-spiking model is $E_{ANN} = 4 \times (2K \times 2K \times 256 + 2K \times 256 \times 256) * (4.6pJ)$.

Our P-SpikeSSM based architecture uses 4 encoder layers, each with number of neurons $N = 256$. The values for $IFR_{in}$ are $[0.08, 0.19, 0.16, 0.17]$, and the corresponding values for $IFR_o$ are $[0.03, 0.12, 0.06, 0.07]$ (corresponding to the four layers). Now, the total computational cost of our SNN considering $N$ neurons in each encoder layer is $E_{SNN} = \sum_{i=1}^{4}(IFR_{in_i} \times 2K \times 2K \times 256 + IFR_{o_i} \times 2K \times 256 \times 256) * (0.9pJ)$. Thus, for ListOps dataset, our SNN model is $36\times$ more energy efficient ($E_{ANN}/E_{SNN}$) than the non-spiking model.

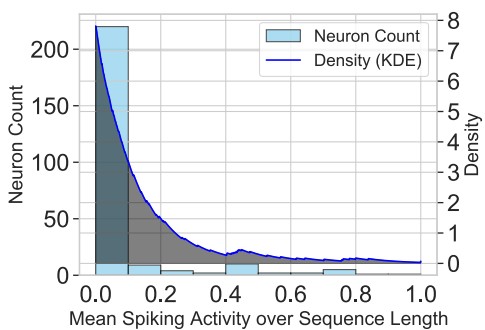

Figure 4: Results obtained after passing randomly sampled inputs from ListOps dataset through our model. Figure consists of histogram representing the count of neurons associated with mean probability of spiking (averaged over sequence length $L$) and Kernel Density Estimation (KDE) plot of the data using an exponential kernel. Thus, over the entire sequence, majority of neurons ($\approx 90\%$) have close to 0 probability of spiking, signifying sparse spiking pattern.

For this analysis, we concentrate exclusively on the cost of arithmetic operations, and did not consider memory I/O transaction costs. Although this methodology does not include architectural details in the energy analysis, it still highlights the computational efficacy of our approach. By leveraging the prevalence of inactive neurons and sparse spiking patterns of active neurons, we can achieve significant improvements in energy and power efficiency on neuromorphic platforms.

## 5  CONCLUSIONS

We propose a computationally efficient probabilistic spiking framework for addressing long-term dependency sequence learning tasks. Instead of using LIF neurons, our model uses the output of P-SpikeSSM neuronal model as the probability for generating spikes using the proposed SpikeSampler layer. To tackle the non-differentiability of this stochastic spiking mechanism, we introduce a surrogate gradient approach, enabling efficient training. To ensure scalability, our architecture features SpikeMixer and ClampFuse layers, enabling effective sequence processing through simplified accumulation-based operations. We evaluate our models on classification tasks involving long-range dependencies, such as the LRA benchmark, ps-MNIST, and the SC10 dataset. Our models consistently outperform transformer-based non-spiking counterparts, achieving state-of-the-art performance among SNN models, while also demonstrating exceptional computational efficiency due to the inherent sparsity of spiking events. To further harness these energy efficiency benefits, future work could explore deploying the model on edge devices and neuromorphic hardware, such as Intel Loihi 2.

ACKNOWLEDGMENTS

This material is based upon work supported in part by the U.S. National Science Foundation under award No. CCSS #2333881, CAREER #2337646, CCF #1955815, and EFRI BRAID #2318101.

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

# A  Deriving Convolutional Representation of P-SpikeSSM Neuronal Dynamics

At time step $t$, the discretized neuronal model exhibits transition dynamics described as follows:

$$h[t] = \overline{A}h[t-1] + \overline{B}x_s[t]$$
$$p_s[t] = \sigma(\overline{C}h[t]) \tag{S1}$$

where, $h[t]$ is the hidden state of the neuron, $p_s[t]$ is the probability of the spiking event $S_t$. $\overline{A}$, $\overline{B}$, $\overline{C}$ are the discretized parameters of the time-invariant system. Considering at time step 0, $h[0] = 0$, we get,

$$h[1] = \overline{B}x_s[1]$$
$$h[2] = \overline{AB}x_s[1] + \overline{B}x_s[2] \tag{S2}$$

Unrolling like this to time step $i$ we get,

$$h[i] = \overline{A}^{i-1}\overline{B}x_s[1] + \overline{A}^{i-2}\overline{B}x_s[2] + \cdots + \overline{AB}x_s[i-1] + \overline{B}x_s[i] = \sum_{j=1}^{i}(\overline{A}^{i-j}\overline{B}x_s[j]) \tag{S3}$$

Thus, the convolutional kernel $\hat{K}$, whose length is given by the length of the input sequence $L$, is defined as,

$$\hat{K} = (\overline{B}, \overline{AB}, \dots, \overline{A}^{L-1}\overline{B}) \tag{S4}$$

Now $H$, i.e., sequence of hidden states can be computed as a non-circular convolution given as, $H = \hat{K} * X_s$, where $X_s$ is the input sequence of spikes.

Thus, $p_s[t] = \sigma((K * X_s)_t) = \sigma(\sum_{j=1}^{t} K_j x_s[t-j+1])$, where $K = (\overline{CB}, \overline{CAB}, \dots, \overline{CA}^{L-1}\overline{B})$.

# B  HiPPO-legS Matrix

HiPPO (high-order polynomial projection operators) (Gu et al., 2020a) is a versatile framework that enables the analysis of various families of measures. Utilizing this operator as either a closed-form ordinary differential equation (ODE) or a linear recurrence, we can efficiently update the optimal polynomial approximation as the input function unfolds over time. HiPPO-legS can generalize to different time scales. HiPPO enables the hidden state to effectively memorize the historical pattern of input spikes (in our paper). The elements of the HiPPO-legS (Scaled Legendre) matrix $\in \mathbb{R}^{n \times n}$ is given below,

$$A_{mk} = -\begin{cases} \sqrt{2m+1}\sqrt{2k+1}, & \text{if } m > k \\ m+1, & \text{if } m = k \\ 0, & \text{if } m < k \end{cases} \tag{S5}$$

## B.1  Computing Kernel $K$

The efficient computation of $K$ has been proposed in literature (Gu et al., 2021a), thus speeding up the parallel training of SSM based neuronal architectures. We briefly go over the overview on how it is achieved. The primary concern in computing $K$ is the repeated multiplication of the state matrix to create the individual terms $K_i$. Thus to compute $K$, the time complexity for a simple approach of chained multiplication is $O(n^2 L)$, where $n$ is the hidden state dimension and $L$ is the sequence length. Now the idea is that, if we had the state matrix to be a diagonal matrix, then theoretically we could compute $K$ efficiently using Vandermonde product. Thus, the goal is to diagonalize the matrix $A$. Now, the ideal scenario is if $A$ is a normal matrix, i.e., it is diagonalizable with a unitary

Table S1: Hyper-parameters used for obtaining the best result on individual datasets used for evaluating P-SpikeSSM-based SNN models.

| | psMNIST | SC10 | ListOps | Text | Retrieval | Image | Pathfinder |
|---|---|---|---|---|---|---|---|
| $K$: #Encoder Layers | 2 | 4 | 4 | 4 | 4 | 6 | 4 |
| $N$: Neurons per Layer | 400 | 256 | 256 | 256 | 200 | 512 | 256 |
| $n$: Hidden State Dim. | 64 | 32 | 16 | 16 | 32 | 32 | 64 |
| $lr$: Learning Rate | 0.01 | 0.002 | 0.005 | 0.0005 | 0.003 | 0.005 | 0.0005 |
| Batch Size | 64 | 32 | 32 | 64 | 32 | 32 | 32 |
| Epochs | 200 | 200 | 200 | 200 | 200 | 200 | 200 |

matrix ($UAU^{-1}$ is a diagonal matrix, where $U$ is a square matrix such that $U^H = U^{-1}$). $A$ is initialized to HiPPO matrices which are not normal matrices. However, HiPPO can be decomposed into a diagonal matrix and a low-rank matrix. Following this, we can leverage previous theoretical results (Gu et al., 2021a) on reducing the underlying SSM to the computation of Cauchy kernels and calculate $K$, in linear order of time complexity w.r.t the sequence length $L$.

## C  DATASET DETAILS

**Permuted Sequential MNIST:** To heighten the complexity of the classification task, permuted sequential MNIST (psMNIST) (Le et al., 2015) reconfigures the presentation of images compared to the original MNIST dataset. While MNIST has each $28 \times 28$ grayscale image as a unified entity, psMNIST arranges the pixels in a sequence and in a permuted order. Thus, tackling this task demands more sophisticated models capable of effectively retaining and synthesizing information over time.

**Speech Command Dataset (SC10):** We use the 10-class subset of Speech Command dataset (Warden, 2018) following previous works (Kidger et al., 2020; Gu et al., 2021a) and evaluate our model on the raw unprocessed signals of length $16000$.

**Long range Arena Benchmark:** To demonstrate the long-range dependency analysis capability of our spiking architecture, we leverage the Long Range Arena (LRA) benchmark (Tay et al., 2020), spanning various classification tasks from textual to image domains. Following are the five tasks utilized for evaluation,

- **ListOps**: In this task, our focus lies in modeling hierarchically structured data within a long-context framework. The sequence length for this task is upto $2K$.

- **Text**: In this task, we process the IMDB dataset (Maas et al., 2011) of movie reviews and perform the task of sentiment analysis in a byte-level. This is done to ensure a long sequence length of $4K$.

- **Retrieval**: In this task, we assess the model's capacity to encode and retain compressed representations essential for matching and retrieval purposes. The input consists of byte-level sequences (of length $4K$) from two documents, and the goal is to analyze their similarity.

- **Image**: In this task, we perform an image classification task based on a sequence of pixels of the original image. The dataset is CIFAR-10 and the sequence length is $1K$.

- **Pathfinder**: In this task, we treat a $32 \times 32$ image as a sequence of pixels of length $1K$. Our objective is to make a binary decision regarding whether two points, depicted as circles, are linked by a path composed of dashes.

## D  ADDITIONAL EXPERIMENTAL RESULTS

In Table S1, we list the optimal set of hyper-parameters used for each of the tasks. We have used a cosine annealing learning rate scheduler with a warmup phase. As mentioned in the main text, the state matrix $A$ is initialized to the HiPPO-legS matrix as shown in Eqn. S5. The step size for discretization ($\Delta$) is restricted between $[0.001, 0.1]$. The memory footprint ranges from $\approx 6GB$ for psMNIST to $\approx 23GB$ for SC10. The dataset splits are aligned with prior literature (Tay et al., 2020).

**Additional Experiments:** We also tested our model on the sequential CIFAR-10 dataset, achieving an accuracy of 85.6%. The hyper-parameters used are similar to the one used for LRA Image dataset, as shown in Table S1. If we pass real valued probabilities instead of spikes, we achieve improved accuracy (87.2%). However, this approach sacrifices the energy efficiency advantages provided by the spiking sparsity in our original model. Our model processes the input as a sequence of pixels, yet it achieves performance comparable to other state-of-the-art models, such as PSN Fang et al. (2024) (PSN: 88.45% and Sliding PSN: 86.70%), which processes the image as a sequence of columns.

**Memory Footprint:** Based on previous analysis (Tay et al., 2020) on LRA tasks, the Transformer models in Table 2 are configured with $4$ layers, hidden dimension of $256$ and $4$ attention heads, resulting in approximately $600K$ parameters in total. Our models establish state-of-the-art performance in the spiking domain and comprehensively outperforms non-spiking transformer based architectures as shown in Table 2. Furthermore, considering identical parameters $(\overline{A}, \overline{B})$ for neurons in the same layer, the average parameter count of our models on LRA ListOps dataset is around $\approx 280K$, representing a reduction of $\approx 2.1\times$ compared to the parameter count of the transformers used for comparison.

