# OpenReview forum: "P-SPIKESSM: HARNESSING PROBABILISTIC SPIKING STATE SPACE MODELS FOR LONG-RANGE DEPENDENCY TASKS"
_ICLR.cc/2025/Conference — ICLR 2025 Poster_

### Official Review · Reviewer_EjY8 · 2024-10-31

**Soundness:** 3
**Presentation:** 3
**Contribution:** 3
**Rating:** 8
**Confidence:** 4

**Summary:**

The authors present a new model that combines state space model (SSM) layers with stochastic (Bernouilli) spiking layers. They test it on various long sequence benchmarks.

**Strengths:**

Very good accuracy, outperforming all other spiking proposals and even most non-spiking ones (all except S4).

**Weaknesses:**

* IMHO, the main weakness is that this network is not fully spiking, and it's not clear if it could be implemented on existing neuromorphic chips (e.g., Intel Loihi). The temporal convolution of the SSM (eq  6) could probably be implemented by delays. But the gelu in eq 10, and the non-local normalization (in ClampFused layers) are problematic. I suggest the authors discuss these issues and possible solutions. Also, I think the authors should cast their network as "Partial SNN" in Table 1-3

* IMHO psMNIST is not challenging enough (SOTA is nearly 100%, which may mask differences between approaches); I would recommend trying as well on sequential CIFAR10/100 (much more challenging).

* The authors seem to ignore the sliding PSN neuron (http://arxiv.org/abs/2304.12760), which has many similarities with their model. Both use spikes. Both use temporal convolutions instead of stateful units and for this reason, both avoid BPTT and are parallelizable. A comparison with the PSN (both qualitative and quantitative) would be useful.

Minor points:

* "A is a parameter controlling the evolution" -> "A is a matrix controlling the evolution"

* Fig 1 b: the y dimension is useless here. I would recommend plotting the curve y = p(t) instead of a heat map.

* L184: "since probability p[t] in [0, 1]." I think in continuous time, p is a pdf, so it could be >1

--

POST REBUTTAL:

My main concerns have been addressed. I raised my score to 8.

**Questions:**

* I think the authors could bypass the SpikeSampler layer and send the real-valued probabilities directly to the next layer. Do you confirm? Have you tried? Of course, the computational advantages of spikes would be lost, but I expect an increase in accuracy, and it would be interesting to quantify it.

* Would it be possible to encourage even sparser activity via some additional term in the loss function?

---

> ### Author Response · Authors · 2024-11-21
>
> **Thank you for your insightful questions, suggestions and positive feedback. We have addressed your comments below and  incorporated your suggestions in the revised version of our paper.**
>
> **Comment 1:**
> > IMHO, the main weakness is that this network is not fully spiking, and it's not clear if it could be implemented on existing neuromorphic chips (e.g., Intel Loihi). The temporal convolution of the SSM (eq 6) could probably be implemented by delays. But the gelu in eq 10, and the non-local normalization (in ClampFused layers) are problematic. I suggest the authors discuss these issues and possible solutions. Also, I think the authors should cast their network as "Partial SNN" in Table 1-3
>
> **Response:**
> Thank you for your insightful comment. To further validate our model's biological plausibility, we replaced the GELU activation in Eq. 10 with a hardware friendly ReLU function [1]. The results of this modification have been included in our ablation studies. Notably, this substitution had minimal impact on accuracy; for example, on the LRA Text task, the model achieved 80.4% accuracy with ReLU compared to 81.2% with GELU. While ReLU offers a straightforward alternative to GELU, future work could focus on developing biologically plausible variants of GELU that are compatible with neuromorphic hardware. In the ablation studies we have also performed an experiment by removing normalization during training.
>
> **Comment 2:**
> >IMHO psMNIST is not challenging enough (SOTA is nearly 100%, which may mask differences between approaches); I would recommend trying as well on sequential CIFAR10/100 (much more challenging).
>
> **Response:**
> Thank you for your suggestion. We tested our model on the sequential CIFAR-10 dataset, achieving an accuracy of 85.6%. This result has been added to the Additional Experimental Results section in the Appendix for reference.
>
>  **Comment 3:**
> >The authors seem to ignore the sliding PSN neuron (http://arxiv.org/abs/2304.12760), which has many similarities with their model. Both use spikes. Both use temporal convolutions instead of stateful units and for this reason, both avoid BPTT and are parallelizable. A comparison with the PSN (both qualitative and quantitative) would be useful.
>
> **Response:**
> The research on PSN neurons is indeed fascinating, and we have now included a discussion of this work in the Related Works section of our paper.
>
>  **Comment 4:**
> > I think the authors could bypass the SpikeSampler layer and send the real-valued probabilities directly to the next layer. Do you confirm? Have you tried? Of course, the computational advantages of spikes would be lost, but I expect an increase in accuracy, and it would be interesting to quantify it.
>
> **Response:**
> Thank you for your insightful observation. As you rightly pointed out, directly passing the probabilities does lead to higher overall accuracy. For instance, on the seq-CIFAR10  task, using real-valued probabilities yields an accuracy of 87.2%. However, as you mentioned, this approach sacrifices the energy efficiency advantages provided by the spike sparsity in our original model.
>
>
>  **Comment 5:**
> > Would it be possible to encourage even sparser activity via some additional term in the loss function?
>
> **Response:**
> Thank you for this excellent observation. Although we did not explore this aspect in our paper, incorporating sparsity regularization techniques [2] could be a promising direction to further reduce spiking activity. This approach would enhance the computational efficiency of the model by promoting sparsity.
>
>  **Suggestions**
> >"A is a parameter controlling the evolution" -> "A is a matrix controlling the evolution"
>
> **Response**: We have modified the paper accordingly.
>
> >  the y dimension is useless here. I would recommend plotting the curve y = p(t) instead of a heat map.
>
> **Response**: Thank you for the insightful suggestion. We have updated the figure in our paper accordingly.
>
> References:
>
>  [1] Timcheck, Jonathan, Sumit Bam Shrestha, Daniel Ben Dayan Rubin, Adam Kupryjanow, Garrick Orchard, Lukasz Pindor, Timothy Shea, and Mike Davies. "The intel neuromorphic DNS challenge." Neuromorphic Computing and Engineering 3, no. 3 (2023): 034005.
>
>  [2] Yan, Y., Chu, H., Jin, Y., Huan, Y., Zou, Z. and Zheng, L., 2022. Backpropagation with sparsity regularization for spiking neural network learning. Frontiers in Neuroscience, 16, p.760298.

---

> > ### Comment · Reviewer_EjY8 · 2024-11-21
> >
> > * Comment 1
> >
> > So the authors seem to agree that their network is not fully spiking due to GeLU (also pointed out by reviewer Reviewer 2Qpj), among other things.
> > So, as I suggested in my first review, the authors should cast their network as "Partial SNN" in Table 1-3
> >
> > * Comment 3
> >
> > >The research on PSN neurons is indeed fascinating, and we have now included a discussion of this work in the Related Works section of our paper.
> >
> > I can't see this discussion?
> >
> > * Comment 4
> >
> > > As you rightly pointed out, directly passing the probabilities does lead to higher overall accuracy. For instance, on the seq-CIFAR10 task, using real-valued probabilities yields an accuracy of 87.2%
> >
> > I think it's worth reporting this in the paper.
> >
> > * My other comments have been addressed.

---

> > > ### Author Response · Authors · 2024-11-25
> > >
> > > **Comment 1:**
> > > Specifying that our model uses gelu
> > >
> > > **Response:**
> > > Thank you for the valuable suggestion. We have updated our tables and clarified that our model, along with certain spiking baselines such as SpikingLMUFormer, employs the GELU activation function. Additionally, we have noted that one of the baselines, BinaryS4D, utilizes both GELU and floating-point matrix multiplications, which are absent in our model. To distinguish such architectures, we have labeled them as Partial SNNs. Furthermore, in the Ablation Study section, we explored a variant of our model using ReLU instead of GELU, observing only a minimal drop in performance. We avoided labelling SpikingLMUFormer and our approach as Partial SNN in order to stress the point that these approaches avoid floating-point matrix multiplications and thereby preserve the benefits of spiking networks. Please let us know if this addresses your concern.
> > >
> > > **Comment 2**
> > >
> > > Comparison with PSN
> > >
> > > **Response:**
> > > We have discussed PSN in the Related Works section (cited as (Fang et al., 2024)))  and also included a detailed comparison in Appendix D. While our model processes the Sequential CIFAR-10 dataset as a sequence of pixels, PSN processes it as a sequence of columns (of pixels). Despite this difference, our model achieves comparable performance.
> > >
> > > **Comment 3**
> > > Reporting model performance with real-valued probability
> > >
> > > **Response:**
> > > Thank you for this suggestion. We have updated the paper accordingly (Appendix D).

---

> > > > ### Author Response · Authors · 2024-11-25
> > > >
> > > > We truly appreciate your decision to increase your score and are deeply grateful for the positive feedback that has helped us improve our work.

---

### Official Review · Reviewer_2Qpj · 2024-11-03

**Soundness:** 2
**Presentation:** 3
**Contribution:** 2
**Rating:** 5
**Confidence:** 4

**Summary:**

To exploit the energy-saving potential of spiking neural networks (SNNs) and to overcome the bottleneck of SNNs in long-range dependency tasks, this work proposes a scalable probabilistic spiking learning framework based on state space models (SSMs). Unlike the other model that simply concatenates SSMs with the leaky integrate-and-fire (LIF), this work incorporates SSMs into the membrane potential update of spiking neurons, and innovatively proposes P-SpikeSSM in a bid to address the performance issues of LIF and overcome its inability of parallel computations. To build a scalable architecture, this work designs several modules, including SpikeSampler, SpikeMixer, and FuseClamp. With the contribution of these modules, P-SpikeSSM outperforms other SNN models in various long-range dependency tasks. In addition, P-SpikeSSM shows a large improvement compared to its ANN counterpart in energy efficiency analysis.

**Strengths:**

1. The motivation for this work is clear and sound, and the presentation of the methodology is very detailed and easy to follow.
2. P-SpikeSSM integrates the recently high-profile SSM into spiking neural networks from a novel perspective, addressing the problem that the commonly used LIF cannot handle parallel computations. It may provide new insights for future research on the combination of SSM and SNN.
3. By incorporating P-SpikeSSM with some well-designed modules, the proposed model achieves better performance than other spiking networks and some non-spiking networks.

**Weaknesses:**

Major points:
1. P-SpikeSSM is an interesting exploration, but the elements that play a key role in it seem to be just an application of what SSM proposes, such as temporal convolution to enable parallel computation.
2. The network cannot be considered a fully spiking neural network, and is best called a partial SNN because of the use of gelu in SpikeMixer Block (Eq. 10). In addition, the introduction of gelu greatly increases the difficulty of deploying this network on neuromorphic hardware.
3. Considering that the network is a partial SNN, it performs only marginally better than BinaryS4D, which is also a partial SNN, on some of the LRA benchmark tasks, and even worse on others (Table 2). Given that P-SpikeSSM and BinaryS4D are both based on SSM, it's debatable whether their advantages over other transformer-based networks mostly stem from SSM.
4. For the experiment on the Speech Command dataset (Table 3), Spiking LMUFormer achieved 96.12% accuracy on the full 35-category dataset, whereas this work is only validated on a subset of 10 categories and is not compared to some recent advanced baselines [1, 2].

Minor point:
1. There is a comparison of energy consumption with the ANN counterpart, but no comparison of the number of parameters with other baselines, which is an important metric influencing performance.

[1] Zeyu Liu, Gourav Datta, Anni Li, and Peter Anthony Beerel. Lmuformer: Low complexity yet powerful spiking model with legendre memory units. ICLR. 2024.

[2] Alexandre Bittar and Philip N Garner. A surrogate gradient spiking baseline for speech command recognition. Frontiers in Neuroscience. 2022.

**Questions:**

1. The benefits of stochasticity? Would there be better or more consistent performance if probabilistic spike sampling is replaced by spike firing at a fixed or learnable threshold?
2. How can the network be trained with the removal of the surrogate gradient (Table 5)?

---

> ### Author Response · Authors · 2024-11-21
>
> **Thank you for your insightful suggestions and valuable feedback. In this rebuttal, we have addressed your comments.**
>
>  **Comment 1:**
> >P-SpikeSSM is an interesting exploration, but the elements that play a key role in it seem to be just an application of what SSM proposes, such as temporal convolution to enable parallel computation.
>
> **Response:**
> Thank you for this insightful comment. While our approach leverages the dynamics of an underlying SSM, our computational model diverges significantly from prior SSM-based architectures. First, unlike previous SSM-based models that process real-valued data, our SSM-based neuronal model operates on sequences of spikes. This distinction allows for a different processing paradigm. Additionally, while previous typical spiking models simply applied an LIF neuron to the output of an SSM to generate spikes, we introduce a probabilistic mechanism for spike generation based on the underlying SSM-based neuronal model. This leads to several key advantages: (a) it achieves improved performance than the baselines, (b) it enables parallelizable training using a surrogate function while leveraging a probabilistic spike sampling approach, and (c) it results in sparse spiking patterns, yielding significant computational efficiency, as detailed in Section 4.2 and illustrated in Fig. 4.
>
>  **Comment 2:**
> >The network cannot be considered a fully spiking neural network, and is best called a partial SNN because of the use of gelu in SpikeMixer Block (Eq. 10). In addition, the introduction of gelu greatly increases the difficulty of deploying this network on neuromorphic hardware.
>
> **Response:**
> Thank you for highlighting this aspect. In response, we explored replacing the GELU layers with the on-chip-compatible ReLU function [3]. The results of this modification are included in our ablation studies. Notably, the substitution had minimal impact on accuracy; for instance, on the LRA Text task, the model achieved 80.4% accuracy with ReLU, compared to 81.2% with GELU. While ReLU serves as a straightforward alternative to GELU, future work could focus on developing biologically plausible GELU variants that are optimized for compatibility with neuromorphic hardware.
>
> **Comment 3:**
> >Considering that the network is a partial SNN, it performs only marginally better than BinaryS4D, which is also a partial SNN, on some of the LRA benchmark tasks, and even worse on others (Table 2). Given that P-SpikeSSM and BinaryS4D are both based on SSM, it's debatable whether their advantages over other transformer-based networks mostly stem from SSM.
>
> **Response:**
> The primary distinction between our proposed P-SpikeSSM and other SSM based approaches, including the BinaryS4D model, lies in the computational paradigm: our neuronal model's SSM operates on sequences of spikes, unlike the real-valued inputs used in BinaryS4D. This design offers a significant computational advantage, as shown in Eqn. [6], by performing convolution operations on sparse spiking inputs, enabling efficient and parallelizable inference. In our approach, the underlying SSM serves as the core neuronal model, responsible for generating probabilities for spike sampling. This eliminates the need for an additional LIF neuron layer, which is a requirement in BinaryS4D. Following the review comments, replacing GELU with ReLU enables our model to be implemented using operations that are compatible with neuromorphic chips [3].
>
> Moreover, with further hyperparameter optimization, our model achieves improved performance on the LRA Text dataset, thus outperforming BinaryS4D on 4 out of 5 datasets in the LRA benchmark. Additionally, the sparse spiking activity throughout our model, as discussed in Section 4.2, provides substantial computational efficiency. Unlike BinaryS4D, which primarily utilizes spiking inputs  only in the linear layers, our approach uses spikes for communication across all layers thereby not compromising on computational benefits offered by spiking architectures.
>
>  **Comment 4:**
> >For the experiment on the Speech Command dataset (Table 3), Spiking LMUFormer achieved 96.12% accuracy on the full 35-category dataset, whereas this work is only validated on a subset of 10 categories and is not compared to some recent advanced baselines [1, 2].
>
> **Response:**
> Thank you for highlighting this aspect. In response, we ran our model on the full 35-category dataset and achieved an accuracy of 96.23%. This surpasses the performance of [2], which reports 94.51% on the same dataset and that of SpikingLMUFormer [1]. We have included these updated results in the revised paper (Section 4.1).

---

> > ### Author Response · Authors · 2024-11-21
> >
> > **Comment 5**
> > >There is a comparison of energy consumption with the ANN counterpart, but no comparison of the number of parameters with other baselines, which is an important metric influencing performance.
> >
> > **Response:**
> > Thank you for bringing this to our attention. We have added a detailed memory footprint analysis in Appendix D. On average, our models have approximately 250K parameters across all five LRA tasks, achieving a 2.4× reduction compared to the 600K parameters of the baseline transformer architecture. For the 35-category Speech command dataset, our model consists of 627K  parameters compared to 1.69M parameters in SpikingLMUFormer.
> >
> >  **Comment 6**
> > >The benefits of stochasticity? Would there be better or more consistent performance if probabilistic spike sampling is replaced by spike firing at a fixed or learnable threshold?
> >
> > **Response:**
> > This is an insightful question. The primary benefit of stochasticity that we leverage in our model is that it enables parallelizable training and inference without any significant overhead. We implemented a version of our model using conventional LIF neurons to conduct an ablation study. The experimental setup mirrors the one outlined in Section 4.1 for the ps-MNIST dataset and both the models have similar number of parameters. Having trained both the models till convergence of validation loss (for 100 epochs), the LIF-based approach achieves a test accuracy of 97.7% on ps-MNIST, compared to 98.4% attained by our stochastic model.
> >
> > Notably, our model also significantly outperforms the LIF-based approach in training and inference efficiency. Training one epoch  (batch size 64)  takes approximately 1.5 minutes with our model, compared to 6.1 minutes for the LIF-based variant on the ps-MNIST dataset. Similarly, inference on the test set requires only 9 seconds for our model versus 17 seconds for the LIF-based approach. This difference stems primarily from the sequential processing bottleneck inherent in LIF-based models, which also depends on Backpropagation Through Time (BPTT) for training. In contrast, our framework leverages a parallelizable architecture, enabling efficient single-pass training with standard backpropagation.
> >
> > All experiments were conducted on an Nvidia RTX A5000 GPU with 24GB of memory, as specified in Section 4.This comparison with conventional LIF neurons has been included in the Ablation Studies section (Section 4.1.2), specifically in lines 459–470.
> >
> >  **Comment 7**
> > >How can the network be trained with the removal of the surrogate gradient (Table 5)?
> >
> > **Response:**
> > This is a great question. While most complex datasets in the LRA benchmark did not train without the surrogate gradient (as stated in Section 4.1.2), we were able to successfully train the LRA Text dataset using a technique where we pass the backpropagating gradient when the neuron spikes (i.e., when the output is 1), and pass 0 when it does not spike. This approach was implemented in PyTorch using a custom autograd function. We have now included this method in the Additional Experimental Results section in the Appendix for reference.
> >
> > References:
> > [1] Zeyu Liu, Gourav Datta, Anni Li, and Peter Anthony Beerel. LMUFormer: Low Complexity Yet Powerful Spiking Model With Legendre Memory Units. ICLR. 2024.
> > [2] Alexandre Bittar and Philip N Garner. A surrogate gradient spiking baseline for speech command recognition. Frontiers in Neuroscience. 2022.
> > [3] Timcheck, Jonathan, Sumit Bam Shrestha, Daniel Ben Dayan Rubin, Adam Kupryjanow, Garrick Orchard, Lukasz Pindor, Timothy Shea, and Mike Davies. "The Intel neuromorphic DNS challenge." Neuromorphic Computing and Engineering 3, no. 3 (2023): 034005.
> >
> > **Thank you for your review and comments. In light of the revisions that we have now made in response to your comments, we kindly request that you reconsider your rating. We are happy to address any further feedback you may have.**

---

> > > ### Author Response · Authors · 2024-11-25
> > >
> > > Thank you for your thoughtful and constructive feedback. We have carefully revised our paper based on your suggestions. As the discussion period is coming to a close, we kindly request that you reconsider your rating in light of these updates. Please let us know if you have any additional feedback—we would be happy to address it.

---

> > > > ### Author Response · Authors · 2024-11-28
> > > >
> > > > Thank you for your valuable feedback, which has been carefully incorporated into our paper. As the discussion period draws to a close, we kindly request you to reconsider your rating, taking these updates into consideration. If you have any further suggestions or comments, we would be delighted to address them.

---

> > > > > ### Author Response · Authors · 2024-12-01
> > > > >
> > > > > Thank you for your insightful suggestions. We have updated our paper to include the suggested clarifications and additional experimental results. As the discussion period is nearing its end, we kindly request you to reconsider your rating in light of these updates. If you have any further suggestions or comments, we would be more than happy to address them.

---

> ### Comment · Reviewer_2Qpj · 2024-12-02
>
> Thanks for your feedback. These clarifications and additional results have addressed most of my concerns. However, I still think that the performance of the model is mainly due to SSM and I'm not sure how significant the use of spikes is for the development of the research community. Overall, I believe that the work falls short of the acceptance threshold.

---

> > ### Author Response · Authors · 2024-12-04
> >
> > We sincerely appreciate your decision to increase the score. Our proposed framework harnesses the dynamics of an SSM-based neuronal model to probabilistically generate spikes. It incorporates components like SpikeMixer and ClampFuse, which not only enhances performance but also enables computational efficiency, as demonstrated in Sec 4.1.2 and Sec. 4.2. Unlike previous approaches that rely on real-valued data, our model processes and communicates through sequences of spikes, achieving significant computational efficiency (as detailed in Section 4.2). Unlike previous approaches, our model eliminates the need for an additional leaky integrate-and-fire (LIF) neuronal layer for spike generation. This allows us to circumnavigate the sequential bottleneck inherent to conventional LIF based models (Sec. 3.1.4) and delivers the following key advantages: (a) superior performance compared to baseline models, (b) parallelizable training through a surrogate function while employing a probabilistic spike sampling approach, and (c) sparse spiking patterns that further enhance computational efficiency (discussed in Sec. 4.2 and illustrated in Fig. 4.).

---

### Official Review · Reviewer_Fa79 · 2024-11-04

**Soundness:** 3
**Presentation:** 3
**Contribution:** 3
**Rating:** 8
**Confidence:** 3

**Summary:**

This work introduces P-SpikeSSM, a stochastic spiking neural network (SNN) that replaces the conventional scalar LIF neuron membrane potential with an SSM linear dynamical system. The multi-dimensional of the SSM increases the expressivity of each neuron. The SSM inside each neuron is read out through a linear map, the output of which is then clamped between 0 and 1 to be interpreted as a probability. This probability is used to generate spikes stochastically during each timestep according to a Bernoulli distribution.

The inspiration for using a multidimensional SSM instead of a scalar membrane potential in the spiking neurons is the increased temporal-dynamic expressivity afforded by SSMs. The neural SSMs are initialized using HiPPO matrices, inspired by prior SSM work.

The reasoning for introducing a probabilistic spiking model is that it affords convenient parallelizability. The authors use the probability as a surrogate operation to propagate gradients through the probabilistic spiking layers.

The authors show through experiments that the P-SpikeSSM performs better or competitively with transformer and other SSM baselines, and the authors perform ablation studies, showing all components of their model discussed in the paper contribute to task performance.

**Strengths:**

Significance.
Replacing the membrane potential of a spiking neuron with an SSM is a well-motivated modification, and the encouraging results presented in this work suggest this modification could be broadly valuable in efficient neural network research.

Originality.
This is the first work that I’ve seen to replace the membrane potential in and SNN with an SSM.

Quality.
The work includes ablation studies to demonstrate the necessity of each component of the model.

Clarity.
The work builds up the model in a step-by-step manner with sufficient details so that the reader can readily understand the motivation for each step.

**Weaknesses:**

On line 141 and the caption for Figure 2, has it truly been proven to be the case that LIF neurons cannot be parallelized? What is the distinction between a non-linear LIF neuron and a linear LIF neuron? (To me, all LIF neurons are non-linear because they have reset terms.) Do efficient algorithms exist for LIF neurons? E.g., see “EXODUS: Stable and Efficient Training of Spiking Neural Networks” by Bauer et al., or the work SLAYER that they reference?

On line 77, the authors claim the surrogate operation Expection[S_t] is novel. I believe such a surrogate operation has already been introduced. E.g., see “Automatic Differentiation of Programs with Discrete Randomness” by Arya et al. NeurIPS 2022.

In Table 1, it is unclear to me the model size/computation budget of these various models. I could imagine model size/compute budget is a key reason why one model would outperform another, so I would like to clearly understand how this table presents a fair comparison. E.g., could you show me how the P-SpikeSSM model is iso-parameter-count?

On line 102, the authors imply that P-SpikeSSM is a “fully” spiking model. While “fully” spiking is not defined in the literature, one might argue that P-SpikeSSM is not fully spiking because each spiking neuron contains a non-spiking SSM within it. I might suggest dropping the “fully” adjective. On line 415, “fully spiking” is implied to be defined as “not requiring floating point MAC operations,” but I am now confused regarding how P-SpikeSSM is fully spiking even though there is an SSM inside every neuron.

I struggle to understand the “Analysis of Energy Efficiency” section. In particular I do not see how the computational cost of the n SSM state variables inside the N P-SpikeSSM neurons in each layer are accounted for. To me, it seems a factor of n is missing.

I noted in line 810 “Memory Footprint” that the SSM parameters in each neuron in a layer can be shared. Are the parameters of the SSMs shared in your experiments?

**Questions:**

I asked the most salient questions above in the “Weaknesses” section. The questions that follow are more minor.

On line 57, the authors state that long-range dependency tasks are largely unexplored in the spiking domain, but then the authors go on to cite works on SNNs with SSMs in paragraph starting on line 139. What do the authors mean by ‘largely unexplored’? I suppose ‘largely unexplored’ is a subjective assessment, however. I might suggest more consistent language.

I would suggest clarifying that the “Analysis of Energy Efficiency” section is treating parallel inference specifically.

Is there anything that can be said about the hyperparameter selection process used in this work, to help justify that the ablation study experiment results are not an artifact of certain hyperparameter choices?

While probabilistic spiking affords convenient parallelization, is there anything that can be said about the drawbacks of or parallelizable alternatives to probabilistic spiking?

Thank you for this fascinating work.

---

> ### Author Response · Authors · 2024-11-21
>
> **We sincerely thank you for your positive review and constructive feedback. We are thrilled that you appreciated our work. Below, we have provided our responses to your comments.**
>
>  **Comment 1:**
> >On line 141 and the caption for Figure 2, has it truly been proven to be the case that LIF neurons cannot be parallelized? What is the distinction between a non-linear LIF neuron and a linear LIF neuron? (To me, all LIF neurons are non-linear because they have reset terms.) Do efficient algorithms exist for LIF neurons? E.g., see “EXODUS: Stable and Efficient Training of Spiking Neural Networks” by Bauer et al., or the work SLAYER that they reference?
>
> **Response:**
> Thank you for your comment. The stateful nature and reset mechanism of the vanilla LIF neuron pose challenges to parallelization. Some implementations address this by removing the reset mechanism to create a linear version of the LIF neuron. In line with works such as EXODUS (which leverage Implicit Function Theorem), another promising learning framework for LIF neuron based SNNs is implicit differentiation at equilibrium [1], which has shown impressive empirical performance. However, [1] has primarily been explored in vision-based tasks and has not been explored  in the context of complex long-range dependency tasks, such as those in the LRA benchmark. We have added this discussion in the revised paper.
>
> **Comment 2:**
> >On line 77, the authors claim the surrogate operation Expection[S_t] is novel. I believe such a surrogate operation has already been introduced. E.g., see “Automatic Differentiation of Programs with Discrete Randomness” by Arya et al. NeurIPS 2022.
>
> **Response:**
> Thank you for highlighting this interesting work. [2] primarily focuses on stochastic behavior in the domain of automatic differentiation, which is used to construct new programs that compute the derivative of an original program. In contrast, our approach analyzes the stochastic nature of spiking events within the context of SNNs. We specifically leverage Expectation[S_t], where S_t represents the random variable for the spiking event, to create a surrogate function that enables efficient and parallel training.
>
>  **Comment 3:**
> >In Table 1, it is unclear to me the model size/computation budget of these various models. I could imagine model size/compute budget is a key reason why one model would outperform another, so I would like to clearly understand how this table presents a fair comparison. E.g., could you show me how the P-SpikeSSM model is iso-parameter-count?
>
> **Response:**
> Thank you for this comment. We have added a detailed memory footprint analysis in Appendix D. On average, our models have approximately 250K parameters across all five LRA tasks, achieving a 2.4× reduction compared to the 600K parameters of the baseline transformer architectures. Our model also demonstrates substantial improvements over the LIF-based approach in both training time and inference speed, achieving superior efficiency during training and inference. Training one epoch (batch size 64) takes approximately 1.5 minutes with our model, compared to 6.1 minutes for the LIF-based variant on the ps-MNIST dataset. Similarly, inference on the test set requires only 9 seconds for our model versus 17 seconds for the LIF-based approach. This difference stems primarily from the sequential processing bottleneck inherent in LIF-based models, which also depends on Backpropagation Through Time (BPTT) for training. In contrast, our framework leverages a parallelizable architecture, enabling efficient single-pass training with standard backpropagation. All experiments were conducted on an Nvidia RTX A5000 GPU with 24GB of memory, as specified in Section 4.This comparison with conventional LIF neurons has been included in the Ablation Studies section (Section 4.1.2), specifically in lines 459–470.
>
>
>  **Comment 4:**
> >On line 102, the authors imply that P-SpikeSSM is a “fully” spiking model. While “fully” spiking is not defined in the literature, one might argue that P-SpikeSSM is not fully spiking because each spiking neuron contains a non-spiking SSM within it. I might suggest dropping the “fully” adjective. On line 415, “fully spiking” is implied to be defined as “not requiring floating point MAC operations,” but I am now confused regarding how P-SpikeSSM is fully spiking even though there is an SSM inside every neuron.
>
> **Response:**
> Thank you for this suggestion, we have modified our paper accordingly and refrained from using such a terminology. Though we have an SSM inside every neuron, we can represent the internal dynamics as a convolution over sparse input spikes (Eqn. 6) in the parallel convolution based inference phase, which allows us to avoid floating point multiplicative operations.

---

> > ### Author Response · Authors · 2024-11-21
> >
> > **Comment 5:**
> > >I struggle to understand the “Analysis of Energy Efficiency” section. In particular I do not see how the computational cost of the n SSM state variables inside the N P-SpikeSSM neurons in each layer are accounted for. To me, it seems a factor of n is missing.
> >
> > **Response:**
> > This is an insightful question. In the "Analysis of Energy Efficiency" section, we demonstrate parallel inference by representing the transition dynamics of the neuron as a convolution over sparse input spikes (Eqn. 6). Since the kernel K (Eqn. 5) can be precomputed prior to inference, we can cache it, effectively eliminating the factor n during inference. This detail has now been included in the paper for clarity.
> >
> >
> >  **Comment 6:**
> > >I noted in line 810 “Memory Footprint” that the SSM parameters in each neuron in a layer can be shared. Are the parameters of the SSMs shared in your experiments?
> >
> > **Response:**
> >  In our experiments, we observed that if we share the parameters A,B,C among neurons of the same layer, in most of the datasets it achieves similar performance as to when we use different individual values. This goes to show that the neuronal dynamics in the neurons in the same layer follows similar behavior. Furthermore, this substantially reduces the parameter count.
> >
> >  **Comment 7:**
> > >On line 57, the authors state that long-range dependency tasks are largely unexplored in the spiking domain, but then the authors go on to cite works on SNNs with SSMs in paragraph starting on line 139. What do the authors mean by ‘largely unexplored’? I suppose ‘largely unexplored’ is a subjective assessment, however. I might suggest more consistent language.
> >
> > **Response:**
> > Thank you for the suggestion. We have revised the paper accordingly.
> >
> >
> >  **Comment 8:**
> > >I would suggest clarifying that the “Analysis of Energy Efficiency” section is treating parallel inference specifically.
> >
> > **Response:**
> > Thank you for this suggestion. We have revised the paper accordingly.
> >
> >  **Comment 9:**
> > >Is there anything that can be said about the hyperparameter selection process used in this work, to help justify that the ablation study experiment results are not an artifact of certain hyperparameter choices?
> >
> > **Response:**
> > This is an insightful question. For all the experiments, we conducted an extensive grid search to identify the optimal set of hyperparameters. Additionally, all models in the ablation study section are iso-parametric, and grid search was used to determine the optimal hyperparameter configurations for each.
> >
> >
> >  **Comment 10:**
> > >While probabilistic spiking affords convenient parallelization, is there anything that can be said about the drawbacks of or parallelizable alternatives to probabilistic spiking?
> >
> > **Response:**
> > Random number generation, a critical component in probabilistic frameworks, can pose implementation challenges. However, prior work [3] has explored efficient solutions for this on neuromorphic hardware, including chips like Loihi 2.
> >
> > References:
> > [1] Xiao, Mingqing, Qingyan Meng, Zongpeng Zhang, Yisen Wang, and Zhouchen Lin. "Training feedback spiking neural networks by implicit differentiation on the equilibrium state." Advances in neural information processing systems 34 (2021): 14516-14528.
> > [2] Arya, Gaurav, Moritz Schauer, Frank Schäfer, and Christopher Rackauckas. "Automatic differentiation of programs with discrete randomness." Advances in Neural Information Processing Systems 35 (2022): 10435-10447.
> > [3] Pierro, Alessandro, Philipp Stratmann, Gabriel Andres Fonseca Guerra, Sumedh Risbud, Timothy Shea, Ashish Rao Mangalore, and Andreas Wild. Solving QUBO on the Loihi 2 neuromorphic processor. arXiv preprint arXiv:2408.03076, 2024.

---

### Official Review · Reviewer_wNo7 · 2024-11-04

**Soundness:** 2
**Presentation:** 3
**Contribution:** 2
**Rating:** 6
**Confidence:** 4

**Summary:**

The article proposes a framework for learning in long-range sequence tasks. It introduces a series of tactics to address the associated challenges. Firstly, the authors employ a stochastic sampler based on a State-Space Model (SSM) neuronal model in conjunction with a differentiable function. Additionally, the paper provides comparison results to demonstrate the effectiveness of the proposed approach.

**Strengths:**

The paper explores the differences between two neuron models: the conventional Leaky Integrate-and-Fire (LIF) model and a modified version that employs sampling-based techniques. Additionally, the authors present comprehensive comparison results to highlight the effectiveness of their method.

**Weaknesses:**

The authors do not provide strong evidence that the sampling-based technique outperforms the conventional LIF neuronal model. The absence of an input/output step may also hinder the flow of internal dynamics information. I suggest that they compare their architecture using both approaches to clearly demonstrate the benefits of their method. While it is evident that their approach facilitates the parallelization of the algorithm, this does not necessarily imply that the solution can be obtained faster, as more steps might be required to achieve the appropriate spiking rate.

**Questions:**

n/a

---

> ### Author Response · Authors · 2024-11-21
>
> **We sincerely thank you for your positive review and constructive feedback. We have added our response to your comments below.**
>
>  **Comment 1:**
> > The authors do not provide strong evidence that the sampling-based technique outperforms the conventional LIF neuronal model. The absence of an input/output step may also hinder the flow of internal dynamics information. I suggest that they compare their architecture using both approaches to clearly demonstrate the benefits of their method. While it is evident that their approach facilitates the parallelization of the algorithm, this does not necessarily imply that the solution can be obtained faster, as more steps might be required to achieve the appropriate spiking rate.
>
> **Response:**
>  We implemented a version of our model using conventional LIF neurons to conduct an ablation study. The experimental setup mirrors the one outlined in Section 4.1 for the ps-MNIST dataset and both the models have similar number of parameters. Having trained both the models till convergence of validation loss (for 100 epochs), the LIF-based approach achieves a test accuracy of 97.7% on ps-MNIST dataset, compared to 98.4% attained by our stochastic model. For an iso-accuracy convergence comparison, our stochastic model achieves the same 97.7% after only 42 epochs of training whereas, the LIF neuron based model requires 76 epochs.
>
> Notably, our model also significantly outperforms the LIF-based approach in training and inference efficiency. Training one epoch (batch size 64) takes approximately 1.5 minutes with our model, compared to 6.1 minutes for the LIF-based variant on the ps-MNIST dataset. Similarly, inference on the test set requires only 9 seconds for our model versus 17 seconds for the LIF-based approach. This difference stems primarily from the sequential processing bottleneck inherent in LIF-based models, which also depends on Backpropagation Through Time (BPTT) for training. In contrast, our framework leverages a parallelizable architecture, enabling efficient single-pass training with standard backpropagation.
>
> All experiments were conducted on an Nvidia RTX A5000 GPU with 24GB of memory, as specified in Section 4. This comparison with conventional LIF neurons has been included in the Ablation Studies section (Section 4.1.2), specifically in lines 459–470.

---

### Author Response · Authors · 2024-12-04

We sincerely thank all the reviewers for their constructive feedback and valuable suggestions, which have been instrumental in improving our paper.

---

### Meta-Review · Area_Chair_5Vm6 · 2024-12-23

**Metareview:**

This paper proposes a probabilistic state space model with spiking neurons that is capable of capturing long timescale dependencies.  The reviewers were impressed by the novelty and clarity of the paper, and by the model's performance on long sequence benchmarks. The main concern they raised was about whether the model can be truly considered a fully spiking neural network. Overall, the paper makes a valuable contribution and I am pleased to report that it has been accepted to ICLR.  Congratulations! Please revise the manuscript to address all points raised during the review process.

**Additional Comments On Reviewer Discussion:**

Reviewers were generally very favorably impressed by this paper, and the discussion mainly focused on clarifying some technical issues about the model itself and how it relates to previous models.  One reviewer (2Qpj) raised the concern that the model is not fully spiking, and also felt that the performance improvements were large enough to warrant acceptance, but I was ultimately persuaded by the more positive appraisals of the other three reviewers.

---

### Decision · Program_Chairs · 2025-01-22

Accept (Poster)